# Improved Efficiency of Perovskite Solar Cells by the Interfacial Modification of the Active Layer

**DOI:** 10.3390/nano9020204

**Published:** 2019-02-05

**Authors:** Zhen Lu, Shangzhi Wang, Huijun Liu, Feng Feng, Wenhua Li

**Affiliations:** 1College of Chemistry and Environmental Engineering, ShanXi DaTong University, Datong 037009, China; luzhen0313@aliyun.com (Z.L.); ws_spring@163.com (S.W.); dtdxlhj@163.com (H.L.); 2Beijing Key Laboratory of Energy Conversion and Storage Materials, College of Chemistry, Beijing Normal University, Beijing 100875, China

**Keywords:** Perovskite solar cell, Interfacial modification, Morphology optimization, Performance improvement

## Abstract

As the most promising material for thin-film solar cells nowadays, perovskite shine for its unique optical and electronic properties. Perovskite-based solar cells have already been demonstrated with high efficiencies. However, it is still very challenging to optimize the morphology of perovskite film. In this paper we proposed a smooth and continuous perovskite active layer by treating the poly (3, 4-ethylenedioxythiophene): poly (styrenesulphonate) (PEDOT:PSS) with pre-perovskite deposition and dimethylsulfoxide (DMSO) rinse. The scanning electron microscope (SEM) and atomic force microscope (AFM) images confirmed a perovskite active layer consisting of large crystal grains with less grain boundary area and enhanced crystallinity. The perovskite devices fabricated by this method feature a high power conversion efficiency (PCE) of 11.36% and a short-circuit current (*J_sc_*) of 21.9 mA·cm^−2^.

## 1. Introduction

Since 2012, methylammonium lead halide (MAPbX_3_, where MA is methylammonium CH_3_NH_3_ and X is a halogen) perovskite-based photovoltaic devices have been studied intensively [1,2,3,4,5,6,7,8,9,10,11]. On one hand, these organic-inorganic hybrid compounds act as light absorbers because their small band gap and high extinction coefficients endow the solar cells strong light absorption in a broad region from visible to the near-infrared; on the other hand, the compounds perform as carrier transporters because their excellent crystallinity and long electron-hole diffusion length [12,13,14,15]. Such distinguished advantages have brought up the value of power conversion efficiency (PCE) of planar structure solar cells as high as 22.7% combined with the precise interface engineering and more importantly, optimizing the morphology of perovskite film [16]. As the active layer to absorb photons and generate charge carries, the quality of CH_3_NH_3_PbI_3_ film is crucial for the photovoltaic performance [5,17,18,19,20]. Tremendously experimental and theoretical efforts have been made to develop effective methods for high quality perovskite film. Among them, spin-coating is mostly used in solution-processed perovskite solar cells (PSCs) due to its simplicity and flexibility [21,22]. It normally performs with one-step or two-step successive deposition of the inorganic and organic precursor [23,24]. The quality of fabricated perovskite films depends on the experimental conditions in a large extent. Various strategies including temperature annealing [19,25,26], solvent annealing [27] and the addition of various additives have been reported to optimize the morphology [20,28,29,30]. Moreover, interface engineering has also been shown as an effective method in morphology manipulation [31,32]. It is achieved by either filling the voids on the surfaces of perovskite film or growing large crystalline grains on hole/electron transport layers (HTLs/HELs). For example, Huang et al. realized that the perovskite grains with different aspect ratio deposited on a wide range of non-wetting HTLs and then attained reduced grain boundary area and charge recombination [33]. Consequently, the morphology of the perovskite layer is highly sensitive to the property of adjacent interface as exemplified by poly (3, 4-ethylenedioxythiophene): poly (styrenesulphonate) (PEDOT: PSS), which is extensive applied as a buffer layer in planar perovskite solar cells. Although depositing polar solvent like dimethylformamide (DMF), DMSO and/or perovskite materials on PEDOT:PSS has been elucidated in the previous studies [34], the improvement of perovskite morphology was inconspicuous and they mainly induced enhancement in conductivity of PEDOT:PSS. It is still a pivotal challenge to form continuous, uniform perovskite film.

Therefore, in this work we demonstrate a strategy for optimizing the morphology of perovskite films via combined effect of dimethylsulfoxide (DMSO) and pre-casted perovskite solution. A smooth and continuous perovskite layer was formed upon PEDOT:PSS layer that pre-treated by perovskite solution deposition and subsequent DMSO rinse. This film consists of large crystal grains with less boundary area and enhanced crystallinity which is preferred to reduce the charge recombination. The impressive increase of short-circuit current (*J_sc_*) from 11.2 to 21.9 mA·cm^−2^ greatly promotes the final PCE to be 11.36%.

## 2. Materials and Methods

### 2.1. Materials and Instruments

PbCl_2_ (99.999%), CH_3_NH_3_I and NH_4_Cl were gained from Sigma Aldrich (St. Louis, MO, USA). PEDOT:PSS (Clevios P VP AI 4083) was purchased from H.C. Starck GmbH (Munich, Germany) and used only after filtered with a 0.45 µm PVDF film. N, N-dimethylformamide (DMF, 99.8%), chlorobenzene (CB, 99.8%) and dimethylsulfoxide (DMSO, 99.8%) were obtained from Acros (Innochem co. LTD, Beijing, China).

UV absorption spectra were determined by a PerkinElmer UV-vis spectrometer model Lambda 750 (Beijing, China). X-ray photoelectron spectroscopy (XPS) was obtained on a Kratos AXIS UTRADLD XPS system (Manchester, UK) with a monochromatic Al Kα (1,486.6 eV). The SEM images were acquired with a JEOL 7000F field emission SEM system (Tescan China, Shanghai, China). AFM measurements were acquired with a Digital Instrument Multimode Nanoscope IIIA (Plainview, TX, USA) operating in the tapping mode. The thickness of the PEDOT:PSS film and perovskite layer were collected by a Dektak surface profilometer (Beijing, China). The average contact angles were obtained from three different positions of the same sample by using an OCA20 instrument (Data Physics, Filderstadt, Germany).

### 2.2. Solar cell Fabrication and Characterization

First, the indium-tin-oxide (ITO) substrates was ultrasonicated with detergents, deionized water, acetone, isopropyl alcohol, deionized water for 25 minutes step by step. The conductivity of ITO was 20 Ω/□. The ITO substrates were treated with ammonium water, H_2_O_2_ and deionized water (1:1:5 by volume) at 160 °C for 30 minutes and followed by deionized water washing. A 38 nm thick PEDOT:PSS layer was deposited on the cleaned ITO glass by spin-coated method at 3500 rpm for 30 s and then dried at 150 °C for 15 min in air. Perovskite precursor was prepared by CH_3_NH_3_I and PbI_2_ (1:1 molar ratio) in DMF (400 mg/mL) at 60 °C for 6 h using NH_4_Cl (31.6 mg/mL) as the additive. The perovskite precursor solution was pin-coated on the ITO/PEDOT:PSS glass at the speed of 3000 rpm for 60 s in a N_2_-glove box and then the perovskite film was rinsed with DMSO. Afterwards, the perovskite layer was deposited onto the treated PEDOT:PSS film with a speed of 3000 rpm for 180 s and heated at 50 °C for 5 min. Phenyl-C_71_-butyric acid methyl ester (PC_71_BM, 20 mg/mL in chlorobenzene) was spin-coated on top of the perovskite layer and the top Ag electrode was deposited with 100 nm under 10^−4^ Pa. The device was measured with an effective area 0.04 cm^2^ in nitrogen condition. *J-V* characteristics were obtained with an Agilent B2902A Source Meter (model XES-301S, SAN-EI) by simulating AM 1.5 G illumination at 100 mW/cm and calibrating the intensity with a standard single-crystal Si photovoltaic cell. 

### 2.3. Fabrication and Characterization of SCLC

The space charge limited current (SCLC) devices were prepared with a structure of ITO/PEDOT:PSS/PVSK/PC_71_BM/Au. A thin PEDOT:PSS film was deposited on the substrate of cleaned ITO by spin-coated method at 3500 rpm/s for 30 s and annealed on a hotplate at 150 °C for 15 min. The PEDOT:PSS layer thickness determined by Dektak surface profilometer is about 38 nm. The perovskite layer was deposited onto the treated PEDOT:PSS film at 3000 rpm/s for 180 s and heated at 50 °C for 5 min. A thin layer of PC_71_BM (20 mg/mL in chlorobenzene, 60–70 nm) was then spin deposited on top of the perovskite layer at 3000 rpm for 60 s inside a N_2_-glovebox. The top Au electrode was obtained by evaporating 100 nm of gold through a shadow mask under high vacuum (10^−4^ Pa). The *J-V* characteristics of the device were equipped with an Agilent B2902A Source Meter with the darkness of 0 V to 1.0 V.

## 3. Results and Discussion

The planar heterojunction perovskite device is fabricated by a typically inverted configuration of ITO/PEDOT:PSS (38 nm)/perovskite layer (~230 nm)/PC_71_BM/Ag (100 nm). Specifically, the CH_3_NH_3_PbI_3_ perovskite films were formed by spin coating method which put a perovskite precursor solution on the top of ITO substrates coated by PEDOT:PSS and dry at 100 °C for 3 min. Based on this procedure, more devices were fabricated as well that composed of treated PEDOT:PSS film which refer to that the PEDOT:PSS layer was subjected to pre-perovskite deposition and subsequently rinsed with DMSO (fully treated) or only DMSO rinsed before the fabrication of perovskite layer. All the devices were fabricated under the optimized condition and their photovoltaic properties were investigated. Figure 1a shows the typical current-voltage (*J-V*) curves of devices prepared with the untreated and treated PEDOT:PSS films. And the photovoltaic performances of the devices including the open-circuit voltage (*V_oc_*), *J_sc_*, fill factor (*FF*) and the final PCE values are listed in Table 1. As shown in Table 1, for the pristine devices which just could obtain a relatively low PCE value of 6.51 %. But a slightly increased PCE is observed for the devices with the PEDOT:PSS layer treated with drops of pure DMSO, which can be mainly attributed to the improvement of *J_sc_* from 11.2 to 13.92 mA cm^−2^. For the fully treated devices, the PCE is greatly promoted to be 11.36 % with a *J*_sc_ of 21.9 mA cm^−2^, a *V_oc_* of 0.960 V and a FF of 0.539. It is noticeable that the almost fold increase (74.5% enhancement) of overall PCE is mainly assigned to the almost over-doubled *J*_sc_, although slightly higher *V_oc_* and declined FF are obtained. According to the previous studies [35], *V_oc_* is highly related with the annealing temperature that is 100, 80 and 50 °C, respectively, corresponding to gradually increased *V_oc_* for pristine (0.840 V), DMSO-treated (0.890 V) and fully treated (0.960 V) devices built in the optimal conditions. Therefore, it is reasonable to understand the higher *V_oc_* presented finally. For FF decreased from 0.70 to 0.539, it is undesirable but indeed observed which will be tentatively explained below.

As the most effective photovoltaic parameter that influences PCE characteristics of perovskite devices, *J*_sc_ is mainly determined by absorption characteristics, exciton dissociation, charge carrier transport/collection efficiency and so forth. Most of these physical processes are closely depending on the quality of active layer, including the morphology and surface coverage. In our case, the dramatically improved *J*_sc_ from 11.2 to 21.9 mA cm^−2^ was achieved for the fully treated devices. To verify these *J*_sc_ values, external quantum efficiencies (EQEs) measurement were carried out. As seen from Figure 1b, both pristine and fully treated devices exhibit a broad spectrum response from 350 nm to 800 nm and the EQE response is much higher after the treatment of pre-perovskite deposition and DMSO rinse. In addition, *J*_sc_ values calculated from integrated EQE curves fit well with that obtained from the *J-V* measurement. As current hysteresis may drastically affect the conclusions regarding the PCE [36,37], we have measured the *J-V* curves of PSCs with DMSO-treated PEDOT:PSS layer (PVSK/DMSO/PVSK) under different scan directions, with a delay time of 20 ms and a voltage step of 0.02 V s^−1^. Very close PCE values of 10.33% in the forward and 9.70% in the reverse scanning direction are presented as telling from Appendix A, suggesting that the PSCs with DMSO-treated PEDOT:PSS layer showed small hysteresis.

Considering that *J*_sc_ is closely correlated to the quality of active layer, the SEM and AFM were carried out to investigate the morphology of perovskite films. It can be told from the SEM images (Figure 2a–d) that both the perovskite films exhibit unique crystalline features and completely “cover” the bottom PEDOT:PSS underlayer. In general, a great deal of domains with length scales in the order of several hundred nanometres consist of these films, meanwhile typical cracks among these crystalline domains are present as well. However, the difference of the CH_3_NH_3_PbI_3_ films deposited on pristine PEDOT:PSS and treated PEDOT:PSS underlayer is extremely evident. In the pristine film (Figure 2a,b), both the crystals with small and large grain size are distributed randomly and large boundaries area is created. Such broad size distribution leads to the coarse surface formation as reflected by the AFM images in Appendix A. It exhibits a root-mean-square (RMS) roughness of 5.41 nm that is in sharp contrast with RMS of 2.64 nm for the film grown on treated PEDOT:PSS underlayer.

Clearly, CH_3_NH_3_PbI_3_ film grown on the treated PEDOT:PSS underlayer demonstrated improved crystalline properties with large grain size and good uniformity, resulting in reduced grain boundary area. Large grain size with less grain boundaries are always pursued because it has a critical influence on photovoltaic parameters of *FF* and *J_sc_*. The main positive effects can lie in the dramatically reduced charge recombination according to the previous reports [25,33]. Hence, a continuous and smooth CH_3_NH_3_PbI_3_ film was formed with the PEDOT:PSS layer treated via pre-perovskite deposition and DMSO rinse. The effect of these treatment and mechanism behind can be described in terms of the removal of PSSH chains and the conformational change of the PEDOT chains as reported previously [34]. Indeed, it was proven in our work again and the decreased amount of PSS was detected by X-ray photoelectron spectroscopy (XPS) as demonstrated in Appendix A. Since PSS (or PSSH) is hydrophilic, its removal reasonably led to an increased contact angle from 12.1^o^ to 45.5^o^ for the fully treated PEDOT:PSS layer (see Figure 3). The contact angle represented the wetting capability of PEDOT:PSS layer to water or N,N-dimethylformamide (DMF) due to its similar dissolubility in these two solvents. The decreased wetting capability is beneficial to the perovskite film growth with large crystalline grain and less boundaries as revealed in the previous studies [33]. This agrees well with our results derived from SEM and X-ray diffraction (XRD) patterns in Figure 4a. The higher intensity of diffraction peaks illustrates enhanced perovskite crystallinity.

To better understand and evaluate the enhanced *J_sc_,* we also performed UV-vis absorption measurements. In Figure 4b, an increased optical absorbance for active perovskite layer deposited on treated PEDDOT:PSS underlayer are observed across the entire absorption region. This is mainly due to the improved crystal growth upon PEDDOT:PSS layer with the treatment of pre-perovskite deposition and DMSO rinse. Moreover, the effect of these treatments can be elucidated from the charge transport point of view. Hence, we measured the hole mobility of perovskite devices by space charge limited current (SCLC) method and the hole-only device was constructed with a structure of ITO/PEDOT:PSS/PVSK/PCBM/Au. The transfer curves are depicted in Appendix A and hole mobilities are calculated based on the literature dates [38,39]. The hole mobility of fully treated devices reached 2.32 × 10^−2^ cm^2^ V^−1^s^-1^ (as shown in Table 2), showing two orders of magnitude higher than that of plain devices, which implied that the charge transport benefited from the modified of the morphology, namely, the formation of much more uniform and smoother film. This boosted hole mobility also coincided well with declined series resistances (*R_s_*, from 6.80 to 4.84 Ω cm^−2^ as shown in Table 2) of perovskite devices with fully treatment that rationalizes the saliently increased *J_sc_*. However, we have to say that meanwhile shunt resistances (*R_sh_*) got decreased unfortunately as shown in Table 2 that brought up the depressed *FF* as mentioned above [40]. Based on the analysis of contact angle, these can be attributed to the weakened interfacial contact between perovskite active layer and PEDOT:PSS layer since the pre-perovskite deposition on PEDOT:PSS layer and subsequent rinse with DMSO induced a significant decrease of hydrophilicity as we discussed in the context.

## 4. Conclusions

In this study, a smooth and continuous perovskite film has been fabricated by pre-perovskite deposition on PEDOT:PSS layer and subsequent DMSO rinse. This film consists of large crystal grains with less grain boundary area and enhanced crystallinity, which can reduce the charge recombination. It was demonstrated that the performance of perovskite devices based on this layer was significantly enhanced and the improved morphology of perovskite film mainly promoted an impressive increasing of *J_sc_* from 11.2 to 21.9 mA cm^−2^. On one hand, the increased optical absorbance for active perovskite layer deposited on treated PEDDOT:PSS underlayer is responsible for the improved *J_sc_*; On the other hand, the highly increased hole mobility of active perovskite layer grown upon treated PEDDOT:PSS layer indicates that the improved charge transport properties made major contribution to the enhancement of *J_sc_* and resulted in a final PCE of 11.36%. In general, it is a simple and flexible method to achieve uniform perovskite film with the treatment of pre-perovskite deposition and subsequent DMSO rinse.

## Figures and Tables

**Figure 1 nanomaterials-09-00204-f001:**
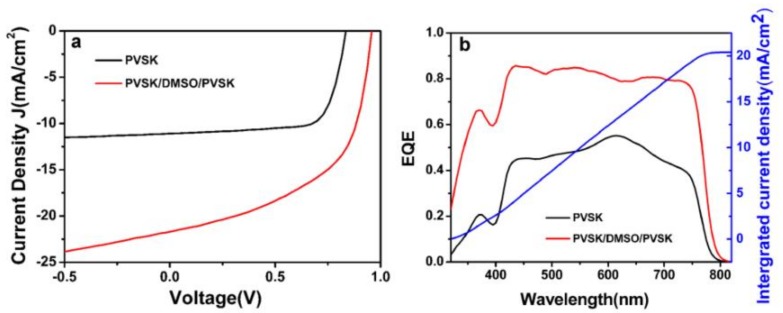
(**a**) *J-V* characteristics and (**b**) EQE spectra of the PSCs on a pristine PEDOT:PSS film (PVSK) (dark) and PEDOT:PSS films treated with perovskite layer rinsed away with DMSO (PVSK/DMSO/PVSK) (red) and photocurrent density (blue) calculated by integrating the EQE with the AM 1.5G solar spectrum.

**Figure 2 nanomaterials-09-00204-f002:**
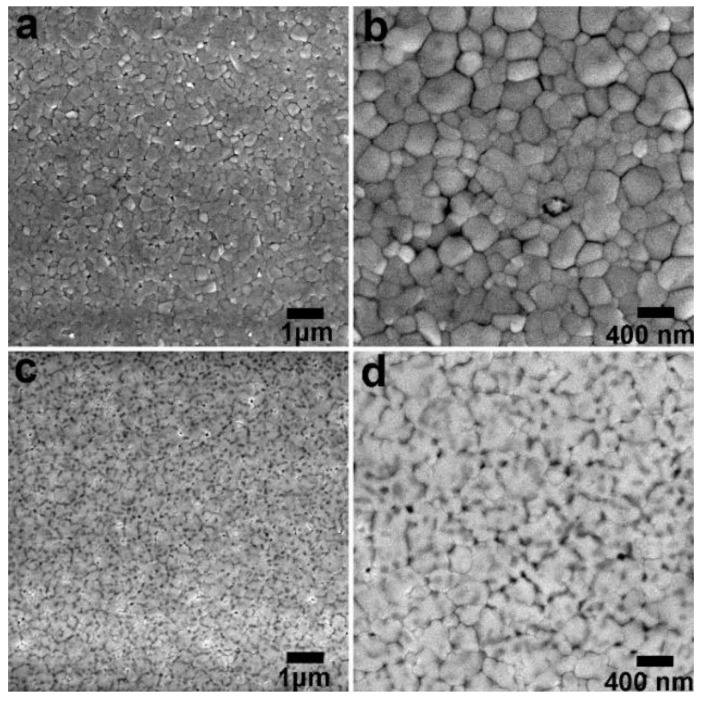
SEM images of perovskite film deposited on (**a**,**b**) perovskite film processed on a pristine PEDOT:PSS film; (**c**,**d**) perovskite film processed on PEDOT:PSS film with the perovskite layer rinsed away by DMSO with high magnification, respectively.

**Figure 3 nanomaterials-09-00204-f003:**
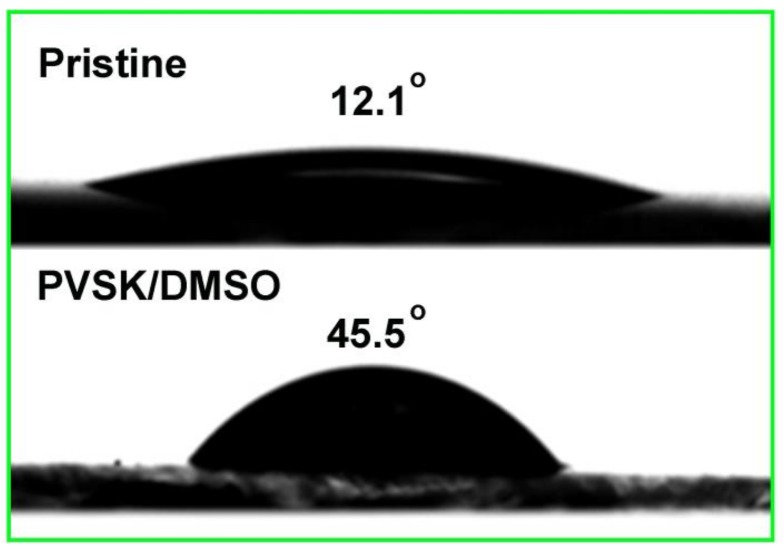
Contact angles with deionized (DI) water drops on the different substrate a pristine PEDOT:PSS film (12.1^o^) and PEDOT:PSS films treated with perovskite layer rinsed away with DMSO (45.5^o^) (PVSK/DMSO).

**Figure 4 nanomaterials-09-00204-f004:**
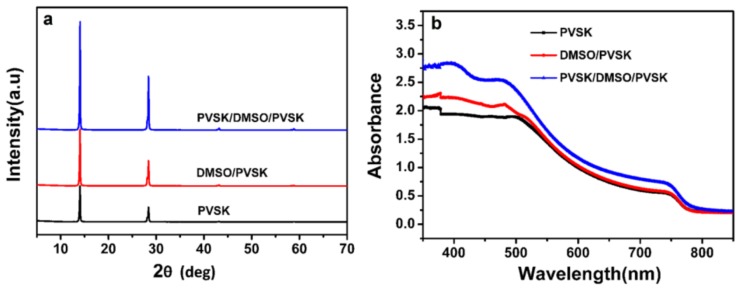
XRD patterns (**a**) and UV absorption spectra (**b**) of perovskite film on a pristine PEDOT:PSS film and PEDOT:PSS films rinsed with DMSO and the pre-deposited perovskite layer rinsed with DMSO.

**Table 1 nanomaterials-09-00204-t001:** Performance data of PSCs on a pristine PEDOT:PSS film (PVSK) and PEDOT:PSS films treated with DMSO (DMSO/PVSK) and perovskite layer rinsed away with DMSO(PVSK/DMSO/PVSK).

Conditions	Annealing temperature (°C)	Time (min)	Voc (V)	Jsc (mA·cm^−2^)	FF	PCE (%)
PVSK	100	3	0.84	11.2	0.692	6.51^a^ (6.37^b^)
PVSK	80	3	0.88	11.2	0.620	6.29^a^ (6.08^b^)
PVSK	50	5	0.97	9.70	0.650	6.14^a^ (5.70^b^)
DMSO/PVSK	80	3	0.89	13.9	0.700	8.28^a^ (7.84^b^)
PVSK/DMSO/PVSK	50	3	0.952	20.7	0.520	10.25^a^ (10.04^b^)
PVSK/DMSO/PVSK	50	5	0.960	21.9	0.539	11.36^a^ (10.95^b^)
PVSK/DMSO/PVSK	50	10	0.950	21.6	0.519	10.68^a^ (10.40^b^)

^a^ The highest PCE among 20 devices; ^b^ Average PCE of 20 devices (Appendix A)

**Table 2 nanomaterials-09-00204-t002:** SCLC data, *R_s_* and *R_Sh_* of perovskite device on a pristine PEDOT:PSS film (PVSK) and PEDOT:PSS films on PEDOT:PSS film with the perovskite layer washed away by DMSO (PVSK/DMSO/PVSK).

Device	*R_s_* (Ω·cm^−2^)	*R_sh_* (Ω·cm^−2^)	SCLC (cm^2^·V^−1^·s^−1^)
PVSK	6.80	679.3	1.66×10^−4^
PVSK/DMSO/PVSK	4.32	100.3	2.32×10^−2^

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
