# Peer review of "Improved Efficiency of Perovskite Solar Cells by the Interfacial Modification of the Active Layer"

_nanomaterials, 2019, doi:10.3390/nano9020204_

Reviewer 1 Report

The authors reported a strategy for optimizing the morphology of perovskite films through combined effect of DMSO and pre-casted perovskite solution on an inverted configuration of ITO/PEDOT:PSS/perovskite layer/PC71BM/Ag. The performance of perovskite devices fabricated by this method gave superior PCE, mainly due to improved Jsc value. The increased value of Jsc was ascribed to the improved morphology of perovskite film. However, on the attached SEM image of perovskite films (Figure 2) it is very difficult to observe any improvement in crystallinity and grain size. Could the authors provide better quality picture, especially for film with the perovskite layer rinsed by DMSO? Moreover, the authors stated that “the difference of the CH3NH3PbI3 films obtained on pristine PEDOT:PSS and treated PEDOT:PSS underlayer is extremely evident.” and I can not find representative SEM image for PEDOT:PSS treated layer neither in main manuscript nor ESI. 

Other comments:

-      Authors stated that Jsc values calculated from integrated EQE curves fit well with that obtained from the J-V measurement. However, on Figure 1b there is no curves corresponding to Jsc values calculated from integrated EQE curves fit.

-      Could the authors explain the different onset observed on EQE and also on UV-vis spectra? Why the UV-vis graph for PEDOT:PSS films treated with perovskite layer rinsed away with DMSO (PVSK/DMSO/PVSK) is redshifted in compared to pristine film in case of using same perovskite CH3NH3PbI3 composition? Is the DMSO change the composition of perovskite layer after rinsing? XPS measurements would be helpful to compare the composition of both perovskite films

I would consider the acceptance of this manuscript in Nanomaterials after regarding the above comments.

Reviewer 2 Report

The authors propose a method for improving the quality of the absorber layer in inverted perovskite solar cells (PSCs). The procedure involves the deposition of CH3NH3PbI3 perovskite films by spin coating onto the PEDOT:PSS layers rinsed with DMSO. The reported PCE increases from 6.51% to 11.36%. The efficiency enhancement is attributed to the increase in photogenerated pairs reflected by the larger short-circuit current, as well as an enhanced carrier mobility determined by SCLC method. AFM analysis shows the increase of perovskite grains by treating PEDOT:PSS layer, which in turn reduce recombinations.

Improving the active layer morphology is certainly a topic of high interest. To support the findings the study combines several techniques in order to asses the morphological and structural properties, such as SEM and AFM, together with (photo)-electrical characterization methods, represented by J-V characteristics, IPCE curves and SCLC.

However, there is strong concern that needs to be addressed before the paper can be considered for publication:

The authors do not discuss at all potential dynamic effects in the J-V characteristics, while it is well known that PSCs typically present hysteresis in the J-V characteristics [Energy Environ. Sci. 8, 995 (2015)]. The information regarding the J-V measurements is incomplete and quite unacceptable for the present-day investigation methods of the PSCs. J-V hysteresis may drastically affect the conclusions regarding the PCE -- see for example [Solar Energy 173, 976 (2018)]. The authors should at least present and compare J-V scans performed in both directions (forward and reverse). There is no information regarding the bias scan rate nor about the scan range that were used. Other alternative procedures that would certify the determined PCE like maximum power point tracking (MPPT) are not present. The simplest way to avoid reporting erroneous PCEs is to perform forward and reverse J-V scans with a small enough scan rate until they match. A discussion regarding the potential dynamic effects is therefore required.

The English language should be revised in the entire manuscript. I would mention only from the title "The Improved Efficience of", which should be changed to "Improved Efficiency of".

As a general remark, the improvement on the PCE, although quite high percentally, it concerns relatively small values (from 6.51% to 11.36%). In this case it is rather unclear if the improvement would stand for higher efficiency PSCs (with PCEs in 15-20% range).  

The statistics on the 20 samples can be added to the Supplementary Material.

Author Response

See attachment.

Round  2

Reviewer 1 Report

I am satisfied with the authors comments and provided corrections. I recommend this paper for publication

Reviewer 2 Report

The authors improved significantly the presentation of their results also in terms of English language, which warrants publication.

I would suggest some minor modifications:  
- in the title:  "Modification of _the_ Active Layer"
- at the end of page 3: "slight hysteresis" --> "small hysteresis"
